# A Secure and Efficient Group Key Agreement Scheme for VANET

**DOI:** 10.3390/s19030482

**Published:** 2019-01-24

**Authors:** Lianhai Liu, Yujue Wang, Jingwei Zhang, Qing Yang

**Affiliations:** 1School of Information Science and Engineering, Central South University, Changsha 410006, China; liulianhai@csu.edu.cn; 2School of Computer Science and Information Security, Guilin University Of Electronic Technology, Guilin 541004, China; gtyqing@hotmail.com; 3Guangxi Key Laboratory of Cryptography and Information Security, Guilin University of Electronic Technology, Guilin 541004, China; yjwang@guet.edu.cn

**Keywords:** vehicular ad hoc network, group key agreement, bilinear maps, batch verification, authentication, shared secret key

## Abstract

A vehicular ad hoc network (VANET) is a special mobile ad hoc network that provides vehicle collaborative security applications using intervehicle communication technology. The method enables vehicles to exchange information (e.g., emergency brake). In VANET, there are many vehicle platoon driving scenes, where vehicles with identical attributes (location, organization, etc.) are organized as a group. However, this organization causes the issue of security threats (message confidentiality, identity privacy, etc.) because of an unsafe wireless communication channel. To protect the security and privacy of group communication, it is necessary to design an effective group key agreement scheme. By negotiating a dynamic session secret key using a fixed roadside unit (RSU), which has stronger computational ability than the on-board unit (OBU) equipped on the vehicle, the designed scheme can help to provide more stable communication performance and speed up the encryption and decryption processes. To effectively implement the anonymous authentication mechanism and authentication efficiency, we use a batch authentication scheme and a shared secret key mechanism among the vehicles, RSUs and trusted authority (TA). We design an efficient group secret key agreement scheme, which satisfies the above communication and security requirements, protects the privacy of vehicles, and traces the real identity of the vehicle at a time when it is necessary. Computational analysis shows that the proposed scheme is secure and more efficient than existing schemes.

## 1. Introduction

A VANET is a special mobile ad hoc network, which is mainly composed of OBUs installed on vehicles and RSUs. The VANET can be used to achieve vehicle-to-vehicle (V2V) communication and vehicle-to-infrastructure (V2I) communication [1,2]. Vehicles in VANET can generate their own driving state information (acceleration, lane changing information, etc.) and collect traffic information such as traffic congestion and slippery road. These information can be used by the vehicles in VANET to improve driving comfort and safety.

Many traffic jams or traffic accidents are caused by the driver’s inability to predict instantaneous and variable road conditions. Intervehicle communication can exchange information about road conditions and vehicle status in time, which helps the driver to determine the driving environment in advance. Thus, effective intervehicle communication can reduce traffic accidents and alleviate traffic congestion. There are three communication modes: unicast, multicast, and broadcast. On some occasions, the emergency message must only be forwarded to specific vehicles. If the message is forwarded by unicast, it takes much time to inform specific vehicles on a road section. If the message is forwarded by broadcast, it is easy to cause a broadcast storm. Multicast is a more efficient method to exchange messages in VANET. However, the wireless channel of VANET may be easily jammed and monitored, which results in data monitoring, tampering and replaying [3]. Some private information of vehicles such as the driver information, driving track, and parking position may be leaked on these channels [4], which would result in potential threats to the drivers and passengers [2]. Therefore, the security of VANET data transmitted on these communication channels must be enhanced.

With the development of group communication application in VANET, the group key agreement mechanism is widely studied. Because of the lower density and high mobility of vehicles [5], the secure and effective group key agreement mechanism for VANET becomes extremely important. Compared to mobile ad hoc networks (MANETs) [6], the vehicles in VANET have lower fixity, communication ability and computing power than the RSU. Therefore, to improve the efficiency of the group key agreement, people can use the advantages of the RSU to complete the computing and communication. The vehicles in VANET may move fast and must dynamically change the running route because there are always vehicles joining or leaving a communication group. A secure secret key agreement scheme must ensure that the legality of new vehicles should be verified before joining the communication group. In addition, when some vehicle leaves the communication group, the RSU can revoke its group key. Therefore, designing a secure and efficient group key agreement for VANET is the key to realize group communication.

In recent years, many studies have been conducted on group key agreement in VANET. A batch authentication scheme [7] was presented to improve the computation efficiency, but the integrity of the request messages were not checked before the batch authentication. Hai [8] proposed an authenticated group key agreement scheme using bilinear pairings, which satisfies all secure group communication requirements for VANET. However, this scheme authenticates vehicles using certificates, which results in low authentication efficiency. Therefore, it is a great challenge to design an efficient and secure group secret key agreement for VANET.

To address the aforementioned problems, we present a group secret key agreement scheme for VANETs with batch verification. Our main contributions are summarized as follows:
The RSU is used at the main node in the group key agreement. Since the RSU has more powerful computing and communication capabilities than the vehicles, the communication cost can be reduced, and the key negotiation efficiency can be increased.The shared secret key mechanism is used among the vehicles, RSUs and TA to effectively implement the anonymous authentication mechanism.The RSU verifies a set of signatures in the batch, which greatly improves the authentication performance compared to individually verification.

The remainder of this paper is organized as follows. We describe the related works in Section 2. The system model and security requirements for VANET are defined in Section 3. We present a secure and efficient group key agreement (SEGKA) scheme in Section 4 and analyze the security and evaluate the performance for SEGKA in Section 5. Finally, the paper is concluded in Section 6.

## 2. Related Work

The VANET provides a series of applications, such as efficiency applications [9,10] (urban traffic management, path planning, etc.), commercial applications [11] (location-based services, path planning etc.), information entertainment applications (video sharing, social networks, etc.), and security applications [12,13,14,15] (rear end warning, road ice testing, etc.). However, the vehicles communicating in the wireless network are easily attacked, particularly when the vehicles have private messages to be shared with others in the same group. Therefore, VANET must build secure channels for group communication. The emergence of group-oriented communication applications has triggered research on group communication security and privacy protection. A difficult problem of group communication is to design an effective group membership authentication key agreement mechanism.

In VANET, the relative position among vehicles may fast and frequently changed, thus the vehicles are often dynamically divided into groups to perform broadcast communication, i.e., group communication. Vehicle group communication refers to communication among vehicles with the same attribute. A secure group communication scheme should be able to ensure that once a new vehicle joins the group and becomes a legitimate group member, it could receive or send messages in the group in time. Also, once some node moves far away, there should be a mechanism to let it leave the current group, so that it cannot continue to enjoy the rights of a legitimate group member and cannot continue to receive or send messages in this group. The group key agreement schemes can be divided into two types: a central node to assign a communication key to other members, and every group member provides a partial key and finally forms a group key. Although the communication mechanism using the asymmetric encryption technology can well satisfy this requirement, it is not suitable for VANET applications because it does not account for the vehicle’s computation capability and complicated key management. To reduce the cost of computing and improve communication efficiency, it is preferable to use symmetrical key encryption in designing VANET communication schemes.

Because of the fast-moving speed and limited communication scope, the secure and effective group key agreement mechanism becomes extremely important. To handle the security and efficiency issues, Han, Hua and Ma [16] proposed a self-authentication and deniable efficient group key agreement (SADEGKA) protocol. The certification efficiency is improved with the group key transmission method without certification authority and prevents the attacker from attacking the legal vehicle through a deniable group key agreement method. However, this scheme is not scalable because every vehicle must verify other vehicles during the key agreement, which increases the verification delay. Chim et al. [17] studied privacy protection and presented a method to verify a batch of signatures within a short time period using two shared secrets. This method enables the existing vehicles to form a group for secure communications. Meanwhile, the RSU is involved in the signature verification process, which greatly mitigates the vehicle’s computing burden. Our scheme also uses shared secrets for the group key agreement to improve the group key agreement efficiency. The RSU always has more powerful computation ability than vehicles; thus, it can speed up the vehicle legality certification. Lei, Yu and Xian [18] proposed an ID-based group authenticated key agreement protocol based on the DBDH assumption and considered the dynamic issue of group communication. In this scheme, the vehicles cannot be anonymous, and their privacy cannot be protected. Zheng et al. [19] introduced an ID-based authenticated group key agreement protocol without the management of certifications. However, both [18,19] have privacy protection problems.

## 3. Preliminaries

In this section, we briefly introduce the system model and bilinear maps. Some notations are shown in Table 1.

### 3.1. System Model

As shown in Figure 1, a typical VANET system includes three types of entities: OBU, RSU and TA.
Each vehicle is equipped with an OBU, which is responsible for the communication with other neighbor OBUs or RSUs using the dedicated short range communication (DSRC) protocol. The OBU has limited computation and storage capabilities. Vehicles periodically broadcast messages of their driving state, e.g., emergency braking. Since the OBU is semicredible, it is necessary for vehicles to sign and authenticate messages transmitted in such unreliable transmission scenarios. Otherwise, the communication channel would be vulnerable to attacks from malicious attackers.RSUs are always distributed on both the roadside and intersections, which are responsible for the vehicle-to-infrastructure communication and infrastructure-to-TA communication. They periodically broadcast the road information (e.g., road congestion) and local environment (e.g., gas station and parking lot) to vehicles to improve the traffic condition. The RSU is also semicredible. The RSU has more powerful computation and communication capability than OBU; thus, it is notably suitable to authenticate vehicles and distribute keys to vehicles, which reduces the authentication latency and improves the communication efficiency [20].TA is a trustworthy third party, which is responsible for the generation and distribution of the private and public keys for OBUs and RSUs. The public key of every entity in the system is certified by a trusted party, so that the corresponding certificate can be publicly verified. This trusted party is also responsible for certificate managements. It also initializes the system and generates system parameters. Before performing the key agreement, OBUs and RSUs must be legally registered in TA. TA allocates the related authentication parameters to RSUs and OBUs. TA can trace the real vehicle identification when it is necessary to realize traceability [21].

### 3.2. Bilinear Maps

Let $G_{1}$ be a cyclic additive group with prime order *q* and $G_{2}$ be a cyclic multiplicative group with the same prime order *q*. The bilinear map is denoted as the mapping $e:G_{1}\times G_{1}\to G_{2}$, which has the following properties:(1)*Bilinearity*$$\begin{array}{c} e\left(aP,bQ\right)=e{\left(P,Q\right)}^{ab} \end{array}$$
where P,Q∈G1,a,b∈Zq*R.(2)*Nondegeneracy*$$\begin{array}{c} e(P,P)\neq 1 \end{array}$$
where *P* is the generator of $G_{1}$.(3)Efficiency

There is an efficient polynomial time algorithm to compute e(P,Q) for any $P,Q\in G_{1}$.

### 3.3. Mathematical Assumptions

**Definition** **1.**
*Discrete Logarithm Problem (DLP): ∀a∈Zq*R, given $P,aP\in G_{1}$, compute a.*


**Definition** **2.**
*Computational Diffie-Hellman Problem (CDHP): ∀a,b∈Zq*R, given $P,aP,bP\in G_{1}$, compute abP.*


**Definition** **3.**
*Decisional Diffie-Hellman Problem (DDHP): ∀a,b,c∈Zq*R, given $P,aP,bP,cP\in G_{1}$, decide whether cP=abP.*


## 4. Scheme Design

To address the security and privacy issues of VANET group communication, this paper designs a secure and efficient group key agreement scheme for VANET in bilinear groups. The vehicles will apply for their own group when registering with the TA. TA will authenticate them and assign relevant group identification according to vehicle attribute. The RSU computes the group key for the vehicles in its coverage according to the group identification. When vehicles are driving in the same RSU coverage area, they with the same group identification initiate group key negotiation due to communication needs. With the relatively fixed location, wide coverage, strong communication and computing capabilities, the RSU is selected as the manager of the group to complete the signature batch authentication of the vehicles and compute and distribute the group key. This can greatly improve the negotiation efficiency of the group key and reduce the communication delay. Figure 2 depicts the process of our group key agreement.
(1)TA initializes the system parameters and sends them to vehicles and RSUs.(2)The RSU requests registration to TA.(3)TA returns the verification information.(4)Vehicle sends the vehicle registration information to TA.(5)TA returns the verification information.(6)TA sends the partial vehicle verification information (the shared key, etc.) to RSUs.(7)Vehicle requests the group key agreement and sends its signature to the RSU.(8)The RSU sends the final group key agreement result to vehicle according to the signature verification.

The scheme contains seven modules, i.e., parameter initialization, vehicle and RSU registration, vehicle signing, RSU verification, group key generation, group member joining, and group member leaving.

### 4.1. Parameter Initialization

TA generates some initial system parameters. This process must only be performed once for the entire system. However, TA may periodically update the system master key to enhance the security performance. The detailed processes are as follows.
(1)TA selects a cyclic additive group $G_{1}$ and a cyclic multiplicative group $G_{2}$ that have bilinear map properties.(2)TA selects a random number s∈Zq*R as the system master key and computes $P_{pub}=sP$ as the corresponding public key.(3)TA selects two cryptographic hash functions: $H:G_{1}\to Z_{q}^{*}$, $h:{\left\{0,1\right\}}^{*}\to Z_{q}^{*}$.(4)TA broadcasts the public parameters $paras=\{G_{1},G_{2},e,q,P,P_{pub},H,h\}$ to all vehicles and RSUs, as shown in Figure 3.

### 4.2. Vehicle and RSU Registration

The vehicle and RSU are registered at TA. TA assigns the corresponding registration information to them, as shown in Figure 4.

TA assigns unique *n*-dimensional column vectors $TID_{i}$, $a_{i}$ and $b_{i}$ to every legitimate vehicle. $TID_{i}$ denotes the vehicle’s real identity, $a_{i}$ is the shared secret between vehicle $V_{i}$ and TA, and $b_{i}$ is the shared secret between vehicle $V_{i}$ and the RSU. TA computes $c_{i}=sH\left(a_{i}⊕TID_{i}\right)$ and sends $REG_{V}=TID_{i}\left|\right|a_{i}\left|\right|b_{i}\left|\right|c_{i}$ to vehicle $V_{i}$ through a secure channel.

TA computes $V_{i}$’s verification $VID_{i}=a_{i}⊕TID_{i}$ and sends $REG_{RSU}=VID_{i}\left|\right|b_{i}$ to the RSU through a secure channel.

### 4.3. Vehicle Signing

In this module, the RSU authenticates the vehicles to prepare for the group key agreement. The detailed processes are described below.

Vehicle $V_{i}$ selects a random nonce $r_{i}$, which is used to prevent an attacker from tracing the vehicle. Then, it generates a pseudo identity $PID_{i}$ that is composed of $PID_{i,1}=r_{i}P$ and $PID_{i,2}\;=\;VID_{i}⊕H\left(b_{i}\cdot PID_{i,1}\right)$

$V_{i}$ calculates the signature $\sigma_{i}=c_{i}+b_{i}c_{i}h\left(M_{i}\right)$, where $M_{i}=PID_{i}\left|\right|T_{i}$ and $T_{i}$ is the signing time. Then, it provides the information $D_{i}=r_{i}||PID_{i}\left|\right|\sigma_{i}\left|\right|T_{i}$ to the RSU through a secure channel, as shown in Figure 5.

### 4.4. RSU Verification

This module enables the RSUs to verify the vehicles’ signatures. The verification can be performed as single verification and batch verification.

(1)Single verification

When the RSU receives the vehicle $V_{i}$’s signature $D_{i}$, it decrypts $D_{i}$ with its secret key $SK_{RSU}$ and checks the freshness of time $T_{i}$. If $T_{i}$ is fresh, that is, $T_{i}$ is within the validity period, the RSU continues to find out $V_{i}$’s verification public key $VID_{i}$ and shared secret key $b_{i}$; then, it verifies whether the received $PID_{i,2}$ is equal to $VID_{i}⊕H\left(b_{i}\cdot PID_{i,1}\right)$. If it is true, the RSU verifies Equation (Equation 1):(1)$$e\left(\sigma_{i},P\right)=e\left(H\left(VID_{i}\right)\left(1+b_{i}h\left(M_{i}\right)\right),P_{pub}\right).$$

Proof of correctness:$$\begin{array}{c} L.H.S\\ =e(c_{i}+b_{i}c_{i}h\left(M_{i}\right),P)\\ =e\left(c_{i},P\right)e\left(b_{i}c_{i}h\left(M_{i}\right),P\right)\\ =e\left(sH\left(a_{i}⊕TID_{i}\right),P\right)e\left(b_{i}sH\left(a_{i}⊕TID_{i}\right)h\left(M_{i}\right),P\right)\\ =e\left(H\left(VID_{i}\right),sP\right)e\left(b_{i}H\left(VID_{i}\right)h\left(M_{i}\right),sP\right)\\ =e\left(H\left(VID_{i}\right),P_{pub}\right)e\left(b_{i}H\left(VID_{i}\right)h\left(M_{i}\right),P_{pub}\right)\\ =e(H\left(VID_{i}\right)\left(1+b_{i}h\left(M_{i}\right)\right),P_{pub})\\ =R.H.S \end{array}$$

Therefore, Equation (Equation 1) holds.

(2)Batch verification

Assume the RSU receives a batch of signatures $D_{1},D_{2},\ldots ,D_{n}$ from vehicles $V_{1},V_{2},\ldots ,V_{n}$. First, the RSU checks the freshness of every time $T_{i}\;\left(1\leq i\leq n\right)$. If all are fresh, then the RSU continues to find the vehicle’s public verification key and the shared secret key and checks whether the second part of the pseudo identity is valid. If all are valid, the RSU verifies the signatures in the batch by checking Equation (Equation 2).
(2)$$e\left(\sum_{i=1}^{n}\sigma_{i},P\right)=e\left(\sum_{i=1}^{n}H\left(VID_{i}\right)\left(1+b_{i}h\left(M_{i}\right)\right),P_{pub}\right).$$

Proof of correctness:$$\begin{array}{c} L.H.S\\ =e(\sum_{i=1}^{n}\left(c_{i}+b_{i}c_{i}h\left(M_{i}\right)\right),P)\\ =e\left(\sum_{i=1}^{n}c_{i},P\right)e\left(\sum_{i=1}^{n}b_{i}c_{i}h\left(M_{i}\right),P\right)\\ =e(\sum_{i=1}^{n}sH\left(a_{i}⊕TID_{i}\right),P)\\ \;\;\;\;e(\sum_{i=1}^{n}b_{i}sH\left(a_{i}⊕TID_{i}\right)h\left(M_{i}\right),P)\\ =e\left(\sum_{i=1}^{n}H\left(VID_{i}\right),sP\right)e\left(\sum_{i=1}^{n}b_{i}H\left(VID_{i}\right)h\left(M_{i}\right),sP\right)\\ =e\left(\sum_{i=1}^{n}H\left(VID_{i}\right),P_{pub}\right)e\left(\sum_{i=1}^{n}b_{i}H\left(VID_{i}\right)h\left(M_{i}\right),P_{pub}\right)\\ =e(\sum_{i=1}^{n}H\left(VID_{i}\right)\left(1+b_{i}h\left(M_{i}\right)\right),P_{pub})\\ =R.H.S \end{array}$$

Therefore, Equation (Equation 2) holds.

### 4.5. Group Key Generation

After the vehicles are authenticated, the RSU generates the group key for the vehicles. The detailed processes are as follows.
(1)The RSU randomly selects a random nonce dRSU∈Zq*R, computes $D_{i}=d_{RSU}PID_{i,1}$, and computes the group key $K_{RSU}$ as follows:
(3)$$K_{RSU}=e\left(\sum_{i=1}^{n}D_{i},d_{RSU}P\right)$$(2)The RSU computes its signature $\sigma_{RSU}=SK_{RSU}H\left(D\right)$, where $D=D_{1}\left|\right|D_{2}\left|\right|\ldots \left|\right|D_{n}$. Then, it broadcasts $Z=\sigma_{RSU}\left|\right|D$ to the vehicles, as shown in Figure 6.(3)After vehicle $V_{i}$ receives *Z*, it first verifies the signature of the RSU by checking Equation (Equation 4) as follows.
(4)$$e\left(\sigma_{RSU},P\right)=e\left(H\left(D\right),PK_{RSU}\right).$$(4)If the signature of the RSU is valid, vehicle $V_{i}$ computes group key $K_{i}$ as follows.
(5)$$K_{i}=e\left(\sum_{i=1}^{n}D_{i},r_{i}^{-1}D_{i}\right)$$

### 4.6. Group Member Joining

Suppose that $V_{1},V_{2},\ldots ,V_{n}$ have a group key as described. The detailed process of a new vehicle $V_{a}$ joining the group is described as follows.
(1)$V_{a}$ selects a random nonce $r_{a}$ and generates a pseudo identity $PID_{a}=\left(PID_{a,1},PID_{a,2}\right)$, where $PID_{a,1}=r_{a}P,PID_{a,2}=VID_{a}⊕H\left(b_{a}\cdot PID_{a,1}\right)$. Then, $V_{a}$ calculates the signature $\sigma_{a}=r_{a}H\left(PID_{a}\right)+b_{a}c_{a}H\left(T_{a}\right)$ and sends $D_{a}=ENC_{PK_{RSU}}(r_{a}||PID_{a}\left|\right|\sigma_{a}\left|\right|T_{a})$ to RSU.(2)When the RSU receives signature $D_{a}$ of vehicle $V_{a}$, it decrypts $D_{a}$ with its secret key $SK_{RSU}$ and checks the freshness of time $T_{a}$. If the time is fresh, the RSU continues to find the public verification key $VID_{a}$ of $V_{a}$ and shared secret key $b_{a}$. The RSU verifies whether the received $PID_{a,2}$ is equal to $VID_{a}⊕H\left(b_{a}\cdot PID_{a,1}\right)$. If it holds, the RSU verifies $V_{a}$’s signature as shown in Equation (Equation 1). If the signature is valid, the RSU allows for vehicle $V_{a}$ to join the group. Then, the RSU reselects a random nonce dRSU′∈Zq*R and recomputes $D_{i}^{\prime }=d_{RSU}^{\prime }PID_{i,1}\left(1\leq i\leq n\right)$ and $D_{a}=d_{RSU}^{\prime }PID_{a,1}$. The RSU computes the group key $K_{RSU}^{\prime }$ as follows.
(6)$$K_{RSU}^{\prime }=e\left(\sum_{i=1}^{n}D_{i}^{\prime }+D_{a},d_{RSU}^{\prime }P\right)$$(3)The RSU computes its signature $\sigma_{RSU}^{\prime }=SK_{RSU}H\left(X^{\prime }\right)$, where $X^{\prime }=D_{1}^{\prime }\left|\right|D_{2}^{\prime }\left|\right|\ldots \left|\right|D_{n}^{\prime }\left|\right|D_{a}$. Then, it broadcasts $Z^{\prime }=\sigma_{RSU}^{\prime }\left|\right|X^{\prime }$ to all vehicles in the group.(4)When vehicles including $V_{a}$ receive $Z^{\prime }$, they verify the signature of the RSU as defined in Equation (Equation 4). If the signature is valid, they compute a new group key $K_{i}^{\prime }$ as follows.
(7)$$K_{i}^{\prime }=e\left(\sum_{i=1}^{n}D_{i}^{\prime }+D_{a},r_{i}^{-1}D_{i}^{\prime }\right)$$

### 4.7. Group Member Leaving

Suppose that $V_{1},V_{2},\ldots ,V_{n}$ have a group key as described above. Let $V_{n}$ be a vehicle leaving the group. The RSU should update the group key for the remaining n−1 vehicles. The detailed process of the group key updating is described as follows.

(1)The RSU selects a random nonce dRSU′∈Zq*R and computes $D_{i}^{\prime }=d_{RSU}^{\prime }PID_{i,1}\;\left(1\leq i\leq n-1\right)$. The RSU computes the group key $K_{RSU}^{\prime }$ as follows.
(8)$$K_{RSU}^{\prime }=e\left(\sum_{i=1}^{n-1}D_{i}^{\prime },d_{RSU}^{\prime }P\right)$$(2)The RSU computes the signature $\sigma_{RSU}^{\prime }=SK_{RSU}H\left(X^{\prime }\right)$, where $D^{\prime }=D_{1}^{\prime }\left|\right|D_{2}^{\prime }\left|\right|\cdot \cdot \cdot \left|\right|D_{n-1}^{\prime }$. Then, it broadcasts $Z^{\prime }=\sigma_{RSU}^{\prime }\left|\right|X^{\prime }$ to the vehicles.(3)After the vehicles receive $Z^{\prime }$, they verify the signature of the RSU as shown in Equation (Equation 4). If the signature is valid, they compute a new group key $K_{i}^{\prime }$ as follows.
(9)$$K_{i}^{\prime }=e\left(\sum_{i=1}^{n-1}D_{i}^{\prime },r_{i}^{-1}D_{i}^{\prime }\right)$$

## 5. Scheme Analysis

In this section, we analyze the correctness, security and performance of our proposed dynamic group key agreement scheme.

### 5.1. Correctness Analysis

Given the group secret keys $K_{i}$ and $K_{j}$ generated by two vehicles $V_{i}$ and $V_{j}$, we have:$$PID_{i,1}=r_{i}P,D_{i}=d_{RSU}PID_{i,1}$$

$$PID_{j,1}=r_{j}P,D_{j}=d_{RSU}PID_{j,1}$$

Thus,

$$r_{i}^{-1}D_{i}=d_{RSU}P$$

$$r_{j}^{-1}D_{j}=d_{RSU}P$$

$$\begin{array}{c} K_{i}\\ =e(\sum_{k=1}^{n}D_{i},r_{i}^{-1}D_{i})\\ =e(\sum_{k=1}^{n}D_{i},d_{RSU}P)\\ =e(\sum_{k=1}^{n}D_{i},r_{j}^{-1}D_{j})\\ =K_{j} \end{array}$$

Therefore, the two keys $K_{i}$ and $K_{j}$ are identical.

### 5.2. Security

In this section, we present detailed analyses on the security and privacy protection of our scheme.

#### 5.2.1. Forward Security

Forward security indicates that even if an attacker can obtain the previous group secret key, it cannot calculate the secret keys of the group in future. In other words, when some vehicle leaves the communication range of the RSU, the RSU will regenerate a random number $d_{RSU}$ for the group key generation and recalculate a secret key from the remaining vehicles $D_{i}$. Thus, the proposed scheme offers forward security.

#### 5.2.2. Backward Security

Backward security indicates that even if the attacker holds the current group key, it cannot calculate the group keys before it joins the group. In other words, before the vehicle enters the communication range of the RSU, it does not hold the previous random number $d_{RSU}$ for the part of the group key, which implies that it cannot calculate the previous group keys. Therefore, the scheme offers the backward security.

#### 5.2.3. Replay Attack Resistance

In the scheme, $D_{i}$ generated by the RSU is different from the secret keys of the vehicles, where a notably strong collision-resistant one-way function *H* is used. Therefore, the group key negotiated is highly independent. In addition, because of the difficulty of CDHP, it is not feasible for any attacker to calculate the secret keys in polynomial time.

#### 5.2.4. Anonymity

$PID_{i}=PID_{i,1}\left|\right|PID_{i,2}$ is a pseudo identity, which contains two random numbers $b_{i}$ and $r_{i}$ generated by TA and the vehicle, respectively. Thus, $PID_{i}$ can well protect the vehicle’s privacy. Since the attacker cannot calculate the true identity $TID_{i}=a_{i}⊕PID_{i,2}⊕H\left(b_{i}\cdot PID_{i,1}\right)$, the proposed scheme supports anonymity.

#### 5.2.5. Traceability

The true identity of the vehicle can only be extracted by TA. Since TA has stored $\left(TID_{i},a_{i},b_{i},VPK_{i}\right)$ during the vehicle registration phase, it can verify the given pseudo identity $PID_{i}=PID_{i,1}\left|\right|PID_{i,2}$. The verification process is shown as follows.
$$\begin{array}{c} a_{i}⊕PID_{i,2}⊕H\left(b_{i}\cdot PID_{i,1}\right)\\ =a_{i}⊕VPK_{i}\\ =TID_{i} \end{array}$$

Therefore, the proposed scheme enables TA to trace the true identities of the vehicles.

#### 5.2.6. Replay Attack Resistance

There is a timestamp $T_{i}$ in the signature generated by vehicle, which enables the RSU to check the freshness of $T_{i}$ to prevent the replay of request for group key generation. Therefore, the proposed scheme can satisfy the replaying resistance.

### 5.3. Performance and Comparison

In this section, the proposed scheme is compared with existing schemes [8,16,22] in terms of computation overhead. Jiang, Zhu and Wang [22] proposed a conditional privacy (ACP) scheme based on anonymized batch authentication in vehicular ad hoc networks. Hai [8] proposed an authenticated group key Agreement (AGKA) scheme for mobile communication based on bilinear. For comparison, only the time-consuming multiplication/division and bilinear pairing operations are considered, and the other efficient operations such as point addition are omitted. Let $T_{par}$ be the execution time of a pairing operation, $T_{mul}$ be the execution time of performing a scale multiplication over an elliptic curve, Terminal be the user terminal node, and ACS be the access control server. The comparison is summarized in Table 2. As shown in Table 2, every procedure of our scheme enjoys constant computing costs, whereas the costs of existing schemes are linear with the group size. With the increase of the number of vehicles, the advantages of our scheme are more and more obvious, that is, the computation costs would not increase. Since both our scheme and ACP use the batch verification method, RSU takes less computations than the other two schemes. Note that OBU in [22] should take *n* multiplications, which requires more computation resources than our scheme. Although the computation cost of OBU in [8] is the same as that of our scheme, there requires a complicated certificate management mechanism, which affects the overall secret key negotiation efficiency.

We conducted experiments on a system with Intel(R) Core(TM) i5-5200U CPU at 2.20 GHz and 8.00 MB memory, using Pairing Based Cryptography Library (PBC) [23]. The elliptic curve is of Type A ($y^{2}=x^{3}+x$), where the element size of group *G* is 256 bits and the size of order *p* is 160 bits. We use Network Simulator 3 (NS3) as communication protocol simulator and follow the IEEE 802.11p standard. Our vehicle mobility model is based on the statistical analysis of the real GPS traces, which includes 360,000 records for a 1043 vehicles network. We extract 50 vehicle traces for delay evaluations. We deployed RSU and TA in the vehicles network. We assume that the OBU, RSU, and TA have completed parameter initialization and registration, and stored related group key negotiation information, such as VID, signature key, etc. The default parameter settings are listed in Table 3

Figure 7 depicts the whole delay in group key negotiation, which includes computation delay and communication delay. In the simulation, we statistically analyzed the average delay of *n* vehicles initiating the negotiation group key. As shown in Figure 7, with the increase of the number of vehicles, the number of channel collisions increases, thereby increasing the communication delay. The communication efficiency of our scheme outperforms other ones, since the computation delay of our scheme is lower than that of other ones. When a new vehicle enters the communication range of RSU to apply for a new session key, the vehicle only needs to send its own group identity and signature to the RSU. Other existing n vehicles do not need to resend their pseudo identities and signatures again. The RSU computes a new group key according to the newly added vehicle information and broadcasts it. In this procedure, only n+2 messages are exchanged. Thus, this phase only requires two rounds to update group key. When a vehicle leaves the communication range of some RSU, the RSU only needs to recalculate the group key based on the information about the remaining n−1 vehicles. In this procedure, only n−1 messages are transferred. This phase only requires one round in updating group key.

## 6. Conclusions

This paper has proposed an authenticated group key agreement scheme for VANET in bilinear groups. The scheme selects the RSU as the main node in group key agreement, adopts the idea of shared secret keys, and realizes the identity authentication of each vehicle. Thorough analyses and comparison demonstrate that the proposed scheme provides privacy protection, traceability and revocability requirements and improves the performance compared to other schemes.

## Figures and Tables

**Figure 1 sensors-19-00482-f001:**
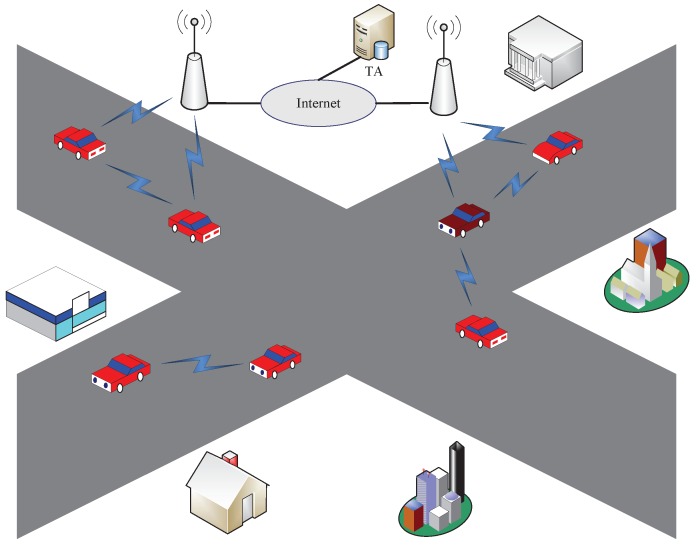
System model.

**Figure 2 sensors-19-00482-f002:**
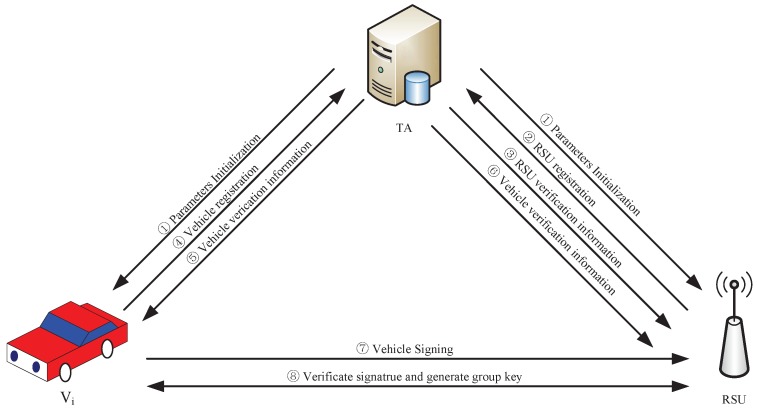
The process of group key agreement.

**Figure 3 sensors-19-00482-f003:**
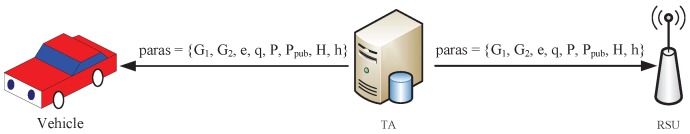
Parameter initialization.

**Figure 4 sensors-19-00482-f004:**
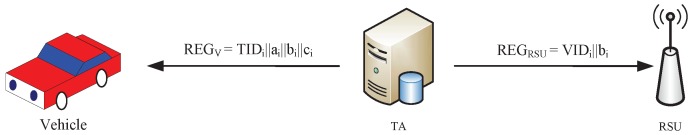
Vehicle and RSU registration.

**Figure 5 sensors-19-00482-f005:**
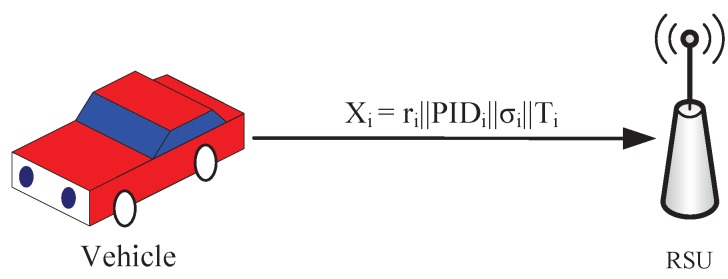
Vehicle signing.

**Figure 6 sensors-19-00482-f006:**
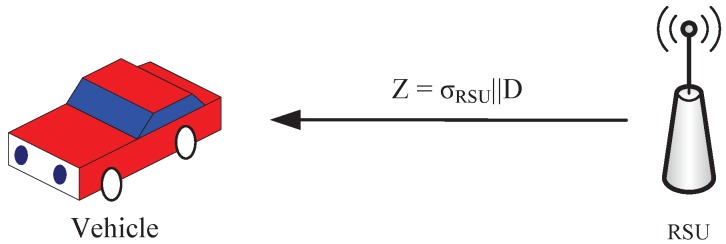
Group key generation.

**Figure 7 sensors-19-00482-f007:**
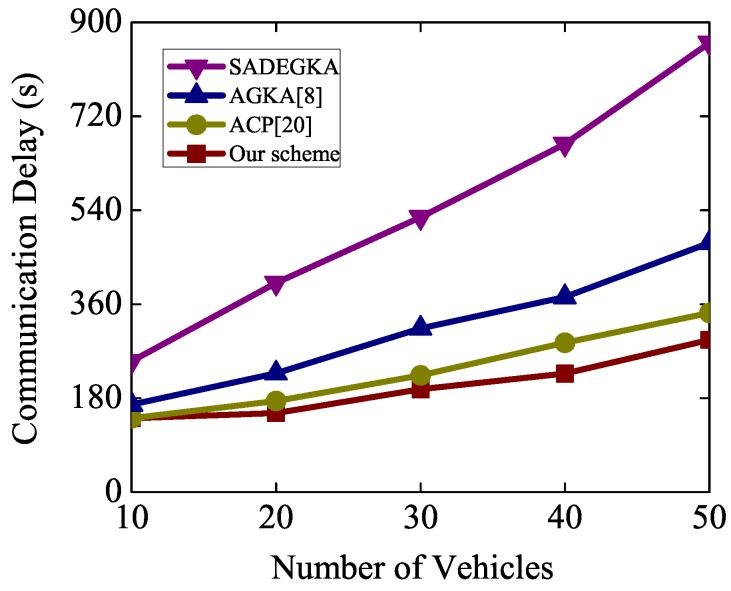
Number of vehicles versus computation delay.

**Table 1 sensors-19-00482-t001:** Notations.

Notations	Definitions
TA	Trusted authority
s	A master key of TA
$P_{pub}$	A public key of TA
$PID_{i}$	The pseudo identity of vehicle
$TID_{i}$	The true identity of vehicle
$V_{i}$	The vehicle number
$VID_{i}$	$V_{i}$’s verification identity
$ENC_{k}\left(M\right)$	Encrypting function of *M* using key *k*
*h*	One secure one-way hash function
*H*	A MapToPoint hash function
$PK_{RSU}$	A public key for the RSU
$SK_{RSU}$	A private key for the RSU
$T_{i}$	The freshness of time
$G_{1}$	The cyclic additive group with prime order *q*
$G_{2}$	The cyclic multiplicative group with the prime order *q*

**Table 2 sensors-19-00482-t002:** Computation Overhead Comparison.

Scheme	OBU/Terminal	RSU/ACS
ACP [22]	$nT_{mul}$	$3T_{par}+\left(2n+1\right)T_{mul}$
AGKA [8]	$3T_{par}+T_{mul}$	$\left(2n-1\right)T_{par}+\left(n+1\right)T_{mul}$
SADEGKA [16]	$2nT_{par}+5nT_{mul}$	$2nT_{par}+4nT_{mul}$
Our scheme	$3T_{par}+T_{mul}$	3$T_{par}+T_{mul}$

**Table 3 sensors-19-00482-t003:** Default parameter settings.

Parameter	Default Value
Vehicles number	50
Communication range	250 (m)
Average speed	40 (kph)
Slot time	$1.3\times 10^{-5}$ (s)

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
