# Peer review of "A Secure and Efficient Group Key Agreement Scheme for VANET"

_sensors, 2019, doi:10.3390/s19030482_

Round 1

Reviewer 1 Report

In this works, authors propose how to apply security mechanisms in group communications focusing on the key agreement in the VANET group communications. However, it is not clear for which kind of groups the proposal is suitable. Is it designed for platooning communications or is it designed for multicasting like applications? It is my personal opinion that more details about applicative scenario can be given. Thus, readers could to better understand how the proposal is applied in the group communications.

In particular, is the proposal performances dependant of group formation, size, and goals? Might undesired behaviors raise if application domain change?

It is clear that the applicative scenario may change some assumptions about System model. When groups are composed of sparse vehicles then several RSUs could be involved in the communications. In particular, in the join and leave process. How does the proposal work in that cases? Reading this point the paper should be improved. I suggest adding a more detailed analysis of possible scenarios because it may help a reader to understand the proposal. Moreover, even if platoons are considered more than one RSU can be involved. In this work, how the RSUs communicate between them concerning the proposed mechanism is not clear.

What about the freshness, the authors mentioned it in section 4.4. This point should be better described is not so easy to understand what is fresh and what is not. Moreover, what happens if vehicles change RSU coverage it does something happening in the management layer? 

In the end, the simulation results section needs to be improved because it lacks important elements such as protocol and network considerations. Since the authors describe a protocol to manage the initialization and runtime behavior of the nodes, then the introduced overhead should be evaluated and analyzed. In particular, it is important to know if it changes the network performances such as scalability and reliability. 

More details are needed to evaluate the goodness of the proposal. Moreover, delays are not considered as well as multihop communications.

Author Response

Response to comments for sensors-394820

We thank your feedback. We have considered these comments carefully and have made some changes accordingly. All changed parts are highlighted in the manuscript.

Comment 1-1:

In this works, authors propose how to apply security mechanisms in group communications focusing on the key agreement in the VANET group communications. However, it is not clear for which kind of groups the proposal is suitable. Is it designed for platooning communications or is it designed for multicasting like applications? It is my personal opinion that more details about applicative scenario can be given. Thus, readers could to better understand how the proposal is applied in the group communications.

Response:

With the development of group communication applications in VANET, the security problems of VANET has attracted more and more attention. For example, an attacker can intercept the communication information of other vehicles such as the vehicle's trajectory, which would compromise the privacy of other vehicles. A malicious attacker may broadcast some wrong traffic information, which may cause traffic accident to some extend, however, the attacker may forge an identity to evade responsibility. Therefore, it is necessary to address the security issues in VANETs. Note that the key negotiation is the underlying core technology to realize secure communication among vehicles in VANETs, which allows vehicles to negotiate a temporary session key for encrypting communication information on an open and unsecure channel.

Our key agreement scheme does not rely on some specific network communication model. In real-world applications, the transcripts in every step of our scheme may be further encoded by some coding algorithm depending on respective network communication model. In Section 2 of the revised manuscript, we have add the following elaboration on application scenario:

In VANET, the relative position among vehicles may fast and frequently changed, thus the vehicles are often dynamically divided into groups to perform broadcast communication, i.e., group communication. A secure group communication scheme should be able to ensure that once a new vehicle joins the group and becomes a legitimate group member, it could receive or send messages in the group in time. Also, once some node moves far away, there should be a mechanism to let it leave the current group, so that it cannot continue to enjoy the rights of a legitimate group member and cannot continue to receive or send messages in this group.

Comment 1-2:

In particular, is the proposal performances dependant of group formation, size, and goals? Might undesired behaviors raise if application domain change?

Response:

In Section5.3 of the revised manuscript, we have add the following elaboration:

As shown in Table 2, every procedure of our scheme enjoys constant computing costs, whereas the costs of existing schemes are linear with the group size. With the increase of the number of vehicles, the advantages of our scheme are more and more obvious, and the computation delay does not increase with the increase of the vehicle.

Comment 1-3:

It is clear that the applicative scenario may change some assumptions about System model. When groups are composed of sparse vehicles then several RSUs could be involved in the communications. In particular, in the join and leave process. How does the proposal work in that cases? Reading this point the paper should be improved. I suggest adding a more detailed analysis of possible scenarios because it may help a reader to understand the proposal. Moreover, even if platoons are considered more than one RSU can be involved. In this work, how the RSUs communicate between them concerning the proposed mechanism is not clear.

Response:

As shown in Sections 4.6 and 4.7, our scheme supports efficient joining and leaving mechanisms for legitimate vehicles. In Section 5.3, we added the performance analysis on these two functionalities.

In applications, as vehicles move forward, they can roam between RSUs. However, each time only one RSU is involved in the group key agreement based on the roaming vehicle information. RSUs do not need to exchange some information when vehicles moving forward.

Comment 1-4:

What about the freshness, the authors mentioned it in section 4.4. This point should be better described is not so easy to understand what is fresh and what is not. Moreover, what happens if vehicles change RSU coverage it does something happening in the management layer?

Response:

In Section 4.4, we added a description of freshness:

If Ti is fresh, that is, Ti is within the validity period,.....

In our manuscript, we focus on addressing the key agreement issue in VANET, which does not rely on some specific network communication model and network hierarchy. In response to comment 1-3, we have mentioned that RSUs are not required to do some computation or exchange some information when vehicles change RSU coverage.

Comment 1-5:

In the end, the simulation results section needs to be improved because it lacks important elements such as protocol and network considerations. Since the authors describe a protocol to manage the initialization and runtime behavior of the nodes, then the introduced overhead should be evaluated and analyzed. In particular, it is important to know if it changes the network performances such as scalability and reliability. More details are needed to evaluate the goodness of the proposal. Moreover, delays are not considered as well as multihop communications.

Response:

In Section 5.3 of this revised version, we have added the performance simulation results and analyses. In this part, we mainly considered the computation costs of our scheme and some related scheme, where Fig. 7 confirms the advantage of our solution compared with existing techniques. Since our scheme does not rely on some specific network topology, we did not simulated the scenario of multihop communication.  

Reviewer 2 Report

The authors propose a group key agreement scheme to be used in VANETs. The proposed scheme is slightly more efficient than existing schemes according to the authors' theoretical analysis. The scheme does seem to implement the required security and privacy goals.

However, the improvement over existing schemes is small, as there is already an existing proposal with the same verification overhead on the OBU side. Moreover, the reliance on RSUs is questionable and the evaluation of the scheme is very brief. For instance a realistic simulative evaluation is missing completely, which would help to verify the feasibility of the scheme and which can be found in all related works that are cited by the authors.

Author Response

Response to comments for sensors-394820

We thank your feedback. We have considered these comments carefully and have made some changes accordingly. All changed parts are highlighted in the manuscript.

Comment 2-1:

The authors propose a group key agreement scheme to be used in VANETs. The proposed scheme is slightly more efficient than existing schemes according to the authors' theoretical analysis. The scheme does seem to implement the required security and privacy goals.

However, the improvement over existing schemes is small, as there is already an existing proposal with the same verification overhead on the OBU side. Moreover, the reliance on RSUs is questionable and the evaluation of the scheme is very brief. For instance a realistic simulative evaluation is missing completely, which would help to verify the feasibility of the scheme and which can be found in all related works that are cited by the authors.

Response:

Indeed, as shown in Table 2, the verification technique in [8] has the same computation complexity as our scheme. However, their scheme [8] takes linear computation costs at the RSU side, while our scheme enjoys constant costs. In this version, please refer to Fig 7 for the simulation results.

Our scheme is designed to support key agreement in a dynamic setting, where vehicles may join and leave the communication group according to their relative distances with some RSU. Here, RSU should be trusted entities, which may maintain all intermediate transcripts at every step and help to trace malicious vehicles if needed. Otherwise, all vehicles that do not trust each other can indeed to perform key agreement, which would bring another issue to trace malicious participants.

In Section 5.3 of this revised version, we have added the performance simulation results and analyses. In this part, we mainly considered the computation costs of our scheme and some related scheme, where Fig. 7 confirms the advantage of our solution compared with existing techniques.  

Reviewer 3 Report

The authors present a group key agreement scheme, based on shared secret and a centralized trusted authority for key distribution in VANETs. The literature review is not extensive, but it is appropriate. The computation overhead is estimated and compared with other approches.

The proposal is interesting, but it requires the deployment of fixed infrastructure and also uses shared secrets for the group key agreement to improve the group key agreement efficiency. In my opinion, this is not suitable for changing environments and high mobility as occuring in VANETs. For this reason, an analysis about the feasibility of these requirements for urban routes and roads/highways should be included, with respect to certificate management. The certificate management when exists a trusted authority would be very simple. So, statements about this should be justified.

Another important issue is the validation of your proposal. The results are obtained from estimations about computation overhead, taking into account the mathematical operations. Nevertheless, a simulation has not been performed. The evaluation of your system, including malicious actors and attacks should be carried out.

Author Response

Response to comments for sensors-394820

We thank your feedback. We have considered these comments carefully and have made some changes accordingly. All changed parts are highlighted in the manuscript.

Comment 3-1:

The authors present a group key agreement scheme, based on shared secret and a centralized trusted authority for key distribution in VANETs. The literature review is not extensive, but it is appropriate. The computation overhead is estimated and compared with other approches.

The proposal is interesting, but it requires the deployment of fixed infrastructure and also uses shared secrets for the group key agreement to improve the group key agreement efficiency. In my opinion, this is not suitable for changing environments and high mobility as occuring in VANETs. For this reason, an analysis about the feasibility of these requirements for urban routes and roads/highways should be included, with respect to certificate management. The certificate management when exists a trusted authority would be very simple. So, statements about this should be justified.

Response:

With the development of group communication applications in VANET, the security problems of VANET has attracted more and more attention. For example, an attacker can intercept the communication information of other vehicles such as the vehicle's trajectory, which would compromise the privacy of other vehicles. A malicious attacker may broadcast some wrong traffic information, which may cause traffic accident to some extend, however, the attacker may forge an identity to evade responsibility. Therefore, it is necessary to address the security issues in VANETs. Note that the key negotiation is the underlying core technology to realize secure communication among vehicles in VANETs, which allows vehicles to negotiate a temporary session key for encrypting communication information on an open and unsecure channel.

Our key agreement scheme does not rely on some specific network communication model. In real-world applications, the transcripts in every step of our scheme may be further encoded by some coding algorithm depending on respective network communication model. In Section 2 of the revised manuscript, we have add the following elaboration on application scenario:

In VANET, the relative position among vehicles may fast and frequently changed, thus the vehicles are often dynamically divided into groups to perform broadcast communication, i.e., group communication. A secure group communication scheme should be able to ensure that once a new vehicle joins the group and becomes a legitimate group member, it could receive or send messages in the group in time. Also, once some node moves far away, there should be a mechanism to let it leave the current group, so that it cannot continue to enjoy the rights of a legitimate group member and cannot continue to receive or send messages in this group.

As shown in Sections 4.6 and 4.7, our scheme supports efficient joining and leaving mechanisms for legitimate vehicles. In Section 5.3, we added the performance analysis on these two functionalities.

In applications, as vehicles move forward, they can roam between RSUs. However, each time only one RSU is involved in the group key agreement based on the roaming vehicle information. RSUs do not need to exchange some information when vehicles moving forward.

As mentioned responses, our scheme provides a cryptographic solution to address the key agreement issue in VANET, which does not rely on some specific network communication model. To support various real-world applications, the transcripts in every step of our scheme may be further encoded by some coding algorithm depending on respective network communication model.

In Section 3.1, we added the following elaboration:

The public key of every entity in the system is certified by a trusted party, so that the corresponding certificate can be publicly verified. This trusted party is also responsible for certificate managements.

Comment 3-2:

Another important issue is the validation of your proposal. The results are obtained from estimations about computation overhead, taking into account the mathematical operations. Nevertheless, a simulation has not been performed. The evaluation of your system, including malicious actors and attacks should be carried out.

Response:

In Section 5.3 of this revised version, we have added the performance simulation results and analyses. In this part, we mainly considered the computation costs of our scheme and some related scheme, where Fig. 7 confirms the advantage of our solution compared with existing techniques. However, the attacker in cryptographic schemes may control some entities, which means it could be able to perform the procedures associated to these entities. Thus, the computation costs of attackers are not separately evaluated. 

Round 2

Reviewer 1 Report

First of all, I would like to say thank you to the authors because they supply answers to all my previous comments. However, I still have some concerns about the results they present. Moreover, It should be better for a reader to know more details about the evaluation campaigns they made for validating their proposal in a VANET environment. What did authors use for performing simulation campaigns? What about parameters authors use for modelling scenarios?  Authors added Figure 7 but what does mean computational delay ( this is evaluated on RSUs or OBUs side ) I am not sure about the trend you have depicted still I have some uncertainties. Because, when the number of vehicles increases, then  RSU should analyse more data to achieve the new key and to update its internal status. Moreover,  I have some concerns about group communications because How groups work is not yet clear.

Moreover, You mentioned that in the group communication only one RSU per group is used, but I am not sure about this assumption. Of course, I think this can be true just in the case of groups composed of close vehicles which belong to the same RSU coverage area.  However, it is crucial for a reader to understand parameters you use for testing your schemes such as the number of vehicles, number of RSU, how RSUs are placed in the area, type of applications and a well-defined policy about group creation and its management.

Moreover, it is not clear who is the manager of the group; is the manager the RSU which covers the area? Or is the manager a vehicle which is moving in the area? In that case, what happens to the group when they change area? 

Author Response

Response to Reviewer 1 Comments

Point 1: What did authors use for performing simulation campaigns?

Response 1:In the simulation, we statistically analyzed the average delay of n vehicles initiating the negotiation group key.

Point 2: What about parameters authors use for modelling scenarios?

Response: We use Network Simulator 3 (NS3) as communication protocol simulator and follow the IEEE 802.11p standard. Our vehicle mobility model is based on the statistical analysis of the real GPS traces, which includes 360,000 records for a 1043 vehicles network. We extract 50 vehicles traces for delay evaluations. We deployed RSU and TA in the vehicles network. We assume that the OBU, RSU, and TA have completed parameter initialization and registration, and stored related group key negotiation information, such as VID, signature key, etc. The default parameter settings are listed in the following table.

Parameter

Default value

Vehicles number

50

Communication range

250 (m)

Average speed

40 (kph)

Slot time

1.3e − 5 (s)

Point 3: Authors added Figure 7 but what does mean computational delay ( this is evaluated on RSUs or OBUs side ) I am not sure about the trend you have depicted still I have some uncertainties. Because, when the number of vehicles increases, then RSU should analyse more data to achieve the new key and to update its internal status.

Response: In the simulation, we statistically analyzed the average delay of n vehicles initiating the negotiation group key. As shown in Fig. 7, with the increase of the number of vehicles, the number of channel collisions increases, thereby increasing the communication delay. The communication efficiency of our scheme outperforms other ones, since the computation delay of our scheme is lower than that of other ones.

Point 4: Moreover, I have some concerns about group communications because How groups work is not yet clear. Moreover, You mentioned that in the group communication only one RSU per group is used, but I am not sure about this assumption. Of course, I think this can be true just in the case of groups composed of close vehicles which belong to the same RSU coverage area.

Response: Vehicle group communication refers to communication among vehicles with the same attribute. These vehicles will apply for their own group when registering with the TA. TA will authenticate them and assign relevant group identification according to vehicle attribute. The RSU computes the group key for the vehicles in its coverage according to the group identification.

Point 5: However, it is crucial for a reader to understand parameters you use for testing your schemes such as the number of vehicles, number of RSU, how RSUs are placed in the area, type of applications and a well-defined policy about group creation and its management.

Response: We use Network Simulator 3 (NS3) as communication protocol simulator and follow the IEEE 802.11p standard. Our vehicle mobility model is based on the statistical analysis of the real GPS traces, which includes 360,000 records for a 1043 vehicles network. We extract 50 vehicles traces for delay evaluations. We deployed RSU and TA in the vehicles network. We assume that the OBU, RSU, and TA have completed parameter initialization and registration, and stored related group key negotiation information, such as VID, signature key, etc. The default parameter settings are listed in the following table.

Parameter

Default value

Vehicles number

50

Communication range

250 (m)

Average speed

40 (kph)

Slot time

1.3e − 5 (s)

Point 6:Moreover, it is not clear who is the manager of the group; is the manager the RSU which covers the area? Or is the manager a vehicle which is moving in the area? In that case, what happens to the group when they change area? 

Response: When vehicles are driving in the same RSU coverage area, they with the same group identification initiate group key negotiation due to communication needs. With the relatively fixed location, wide coverage, strong communication and computing capabilities, the RSU is selected as the manager of the group to complete the signature batch authentication of the vehicles and compute and distribute the group key. This can greatly improve the negotiation efficiency of the group key and reduce the communication delay.

    When a new vehicle enters the communication range of RSU to apply for a new session key, the vehicle only needs to send its own group identity and signature to the RSU. Other existing n vehicles do not need to resend their pseudo identities and signatures again. The RSU computes a new group key according to the newly added vehicle information and broadcasts it. When a vehicle leaves the communication range of some RSU, the RSU only needs to recalculate the group key based on the information about the remaining $n-1$ vehicles. This phase only requires one round in updating the group key.

Reviewer 2 Report

In. their revision, the authors addressed some but not all of my comments sufficiently. The added performance evaluation helps to differentiate the scheme against existing schemes.

However, the evaluation is still missing a proper network simulation to evaluate whether the scheme works well in dynamic VANET scenarios. This should be performed using a network simulator such as ns-3 with realistic simulation of PHY, MAC and network protocols.

Author Response

Point 1: However, the evaluation is still missing a proper network simulation to evaluate whether the scheme works well in dynamic VANET scenarios. This should be performed using a network simulator such as ns-3 with realistic simulation of PHY, MAC and network protocols.

Response: We use Network Simulator 3 (NS3) as communication protocol simulator and follow the IEEE 802.11p standard. Our vehicle mobility model is based on the statistical analysis of the real GPS traces, which includes 360,000 records for a 1043 vehicles network. We extract 50 vehicles traces for delay evaluations. We deployed RSU and TA in the vehicles network. We assume that the OBU, RSU, and TA have completed parameter initialization and registration, and stored related group key negotiation information, such as VID, signature key, etc. The default parameter settings are listed in the following table.

Parameter

Default value

Vehicles number

50

Communication range

250 (m)

Average speed

40 (kph)

Slot time

1.3e − 5 (s)

Round 3

Reviewer 1 Report

The authors provide satisfying answers to my previous questions, and I Better understand the paper and their assumptions. However, I suggest to include the answers they provide also in the paper. 

Author Response

Dear Professor,

Thank you for your suggestions, including current and former, which have important guiding significance for our paper writing and scientific research work.

Yours sincerely,

Our responses to your comments are as following,

Point:The authors provide satisfying answers to my previous questions, and I Better understand the paper and their assumptions. However, I suggest to include the answers they provide also in the paper. 

Response: We have included the answers to your previous questions in the revised manuscript. The modified parts were marked with blue highlights in the revised manuscript.

The answers to your previous questions are as following,

Point 1: What did authors use for performing simulation campaigns?

Response 1: In the simulation, all algorithms are implemented using C language. Our code uses the Pairing Based Cryptography Library (PBC, http://crypto.stanford.edu/pbc/), which has been generally used in simulating crypto-systems. We also use Network Simulator 3 (NS3) as communication protocol simulator to test communication delay.

Point 2: What about parameters authors use for modelling scenarios?

Response 2: In the simulation, the elliptic curve is of Type A (y2 = x3 +x), where the element size of group G is 256 bits and the size of order p is 160 bits.

We use Network Simulator 3 (NS3) as communication protocol simulator and follow the IEEE 802.11p standard. Our vehicle mobility model is based on the statistical analysis of the real GPS traces, which includes 360,000 records for a 1043 vehicles network. We extract 50 vehicles traces for delay evaluations. We deployed RSU and TA in the vehicles network. We assume that the OBU, RSU, and TA have completed parameter initialization and registration, and stored related group key negotiation information, such as VID, signature key, etc. The default parameter settings are listed in the following table.

Parameter

Default value

Vehicles number

50

Communication range

250 (m)

Average speed

40 (kph)

Slot time

1.3e − 5 (s)

Point 3: Authors added Figure 7 but what does mean computational delay ( this is evaluated on RSUs or OBUs side ) I am not sure about the trend you have depicted still I have some uncertainties. Because, when the number of vehicles increases, then RSU should analyse more data to achieve the new key and to update its internal status.

Response 3: Figure 7 depicts the whole delay in group key negotiation, which includes computation delay and communication delay. In the simulation, we statistically analyzed the average delay of n vehicles initiating the negotiation group key. As shown in Figure 7, with the increase of the number of vehicles, the number of channel collisions increases, thereby increasing the communication delay. The communication efficiency of our scheme outperforms other ones, since the computation delay of our scheme is lower than that of other ones.

Point 4: Moreover, I have some concerns about group communications because How groups work is not yet clear. Moreover, You mentioned that in the group communication only one RSU per group is used, but I am not sure about this assumption. Of course, I think this can be true just in the case of groups composed of close vehicles which belong to the same RSU coverage area.

Response 4: Vehicle group communication refers to communication among vehicles with the same attribute. These vehicles will apply for their own group when registering with the TA. TA will authenticate them and assign relevant group identification according to vehicle attribute. The RSU computes the group key for the vehicles in its coverage according to the group identification.

Point 5: However, it is crucial for a reader to understand parameters you use for testing your schemes such as the number of vehicles, number of RSU, how RSUs are placed in the area, type of applications and a well-defined policy about group creation and its management.

Response 5: We use Network Simulator 3 (NS3) as communication protocol simulator and follow the IEEE 802.11p standard. Our vehicle mobility model is based on the statistical analysis of the real GPS traces, which includes 360,000 records for a 1043 vehicles network. We extract 50 vehicles traces for delay evaluations. We deployed RSU and TA in the vehicles network. We assume that the OBU, RSU, and TA have completed parameter initialization and registration, and stored related group key negotiation information, such as VID, signature key, etc. The default parameter settings are listed in the following table.

Parameter

Default value

Vehicles number

50

Communication range

250 (m)

Average speed

40 (kph)

Slot time

1.3e − 5 (s)

Point 6: Moreover, it is not clear who is the manager of the group; is the manager the RSU which covers the area? Or is the manager a vehicle which is moving in the area? In that case, what happens to the group when they change area? 

Response 6: When vehicles are driving in the same RSU coverage area, they with the same group identification initiate group key negotiation due to communication needs. With the relatively fixed location, wide coverage, strong communication and computing capabilities, the RSU is selected as the manager of the group to complete the signature batch authentication of the vehicles and compute and distribute the group key. This can greatly improve the negotiation efficiency of the group key and reduce the communication delay. 

When a new vehicle enters the communication range of RSU to apply for a new session key, the vehicle only needs to send its own group identity and signature to the RSU. Other existing n vehicles do not need to resend their pseudo identities and signatures again. The RSU computes a new group key according to the newly added vehicle information and broadcasts it. When a vehicle leaves the communication range of some RSU, the RSU only needs to recalculate the group key based on the information about the remaining $n-1$ vehicles. This phase only requires one round in updating the group key.

Reviewer 2 Report

In addressing my remaining comments, the authors have included a simulative evaluation of their proposed scheme. I believe the manuscript can be published with the now included evaluation.

Author Response

Dear Professor,

Thank you for your suggestions, including current and former, which have important guiding significance for our paper writing and scientific research work.

Yours sincerely,

Our responses to your comments are as following,

Point:In addressing my remaining comments, the authors have included a simulative evaluation of their proposed scheme. I believe the manuscript can be published with the now included evaluation.

Response: We have improved the paper. The modified parts were marked with blue highlights in the revised manuscript. Thank you for your recognition of the paper.
